# Roles of Brassinosteroids in Plant Reproduction

**DOI:** 10.3390/ijms21030872

**Published:** 2020-01-29

**Authors:** Zicong Li, Yuehui He

**Affiliations:** 1Key Laboratory of Plant Development and Environmental Adaption Biology, Ministry of Education, School of Life Sciences, Shandong University, Qingdao 266237, China; 2National Key Laboratory of Plant Molecular Genetics & Shanghai Center for Plant Stress Biology, Chinese Academy of Sciences Center for Excellence in Molecular Plant Sciences, Shanghai 201602, China; yhhe@sibs.ac.cn

**Keywords:** brassinosteroid, reproduction, signaling pathway, hormone crosstalk, flowering time

## Abstract

Brassinosteroids (BRs) are a group of steroid hormones, essentially important for plant development and growth. BR signaling functions to promote cell expansion and cell division, and plays a role in etiolation and reproduction. As the phytohormone originally identified in the pollen grains of *Brassica napus*, BR promotes the elongation of stigma. Recent studies have revealed that BR is also critical for floral transition, inflorescence stem architecture formation and other aspects of plant reproductive processes. In this review, we focus on the current understanding of BRs in plant reproduction, the spatial and temporal control of BR signaling, and the downstream molecular mechanisms in both the model plant *Arabidopsis* and crops. The crosstalk of BR with environmental factors and other hormones in reproduction will also be discussed.

## 1. Introduction

Brassinosteroid (BR) is an important plant hormone in growth and development. BR is ubiquitously distributed in all growing tissues of higher plants, with a much higher concentration in fruit, seeds and pollen. In *Arabidopsis,* the most active form of BR, brassinolide (BL), is converted from the precursor campesterol (CR). CR is first synthesized to campestanol (CN) and subsequently to castasterone (CS) through two parallel biosynthetic pathways, the early and late C-6 oxidation pathways [1]. The enzyme in charge of the conversion of 4-en-3-one to 3-one and 22-OH-4-en-3-one to 22-OH-3-one is DEETIOLATED 2 (DET2) [2]. While the *C*-22 hydroxylation reaction is mediated by DWARF4 (DWF4), the *C*-23 hydroxylation catalysis is mediated by CONSTITUTIVE PHOTOMORPHOGENESIS AND DWARFISM (CPD) [3,4]. The oxidation of *C*-6 is catalyzed by BR6ox [5]. Two P450 mono-oxygenases, CYP90C1 and CYP90D1, are responsible for the conversion of 22-OH-4-en-3-one, 22-OH-3-one, and 3-epi-6-deoxoCT to 23-hydroxylated products [6].

BR is perceived by a receptor kinase BRASSINOSTEROID INSENSITIVE 1 (BRI1) on the plasma membrane [7,8]. Mutants of *bri1* show a variety of grown defects which are very similar to strong BR deficient mutants, including skotomorphogenesis, extreme dwarfism under light and male infertility. BRI1 is a member of plant-specific giant protein family of serine/threonine leucine-rich repeat receptor-like kinase, which has more than 200 homologs in *Arabidopsis* [9]. The extracellular region of BRI1 consists of 24 LRRs with an interruption of an island domain of methionine-rich repeat. The intracellular region can be further divided into three subdomains: a juxtamembrane region, a canonical S/T kinase and a short C-terminal extension [10]. Three homologs of BRI1 have been characterized in *Arabidopsis* with two have high BL-binding affinity [11,12,13].

After receiving BR, BRI1 resumes kinase activity by recruiting the co-receptor kinase BRI1-ASSOCIATED RECEPTOR KINASE 1 (BAK1) and dissociating from the inhibitory protein BRI1 KINASE INHIBITOR 1 (BKI1) [14,15,16,17]. Then the kinase domains of BRI1 and BAK1 are transphosphorylated and the phosphorylated BKI1 can associate with the 14-3-3 family proteins to further amplify BR signaling [16,18]. Another two plasma membrane-anchored cytoplasmic kinases, BRASSINOSTEROID-SIGNALLING KINASE 1 (BSK1) and CONSTITUTIVE DIFFERENTIAL GROWTH 1 (CDG1) are also phosphorylated by activated BRI1 to inactivate the phosphatase BRI1-SUPPESSOR 1 (BSU1) [19,20,21]. BSU1 in turn dephosphorylates a conserved tyrosine residue of BRASSINOSTEROID INSENSITIVE 2 (BIN2), leading to the inactivation of this GSK3-like kinase [22]. The function of BIN2 is to phosphorylate and inactivate two homologous transcription factors, BRASSINAZOLE RESISTANT 1 (BZR1) and BR1-EMS-SUPPRESSSOR 1 (BES1) in the absence of BR [23,24,25]. The phosphorylation leads to the deactivation of these two transcription factors [26]. In high BR level, BSU1 inactivates BIN2 and unphosphorylated BZR1 and BES1 can initiate BR regulated gene activation and repression [23,27]. BZR1 and BES1 initiate BR responsive gene expression by recognizing and binding to the BR response DNA *cis*-element BR-Response Element (BRRE, CGTGC/TG) and E-box (CANNTG) [23,28]. Gain-of-function *bzr1-1D* and *bes1-D* contain a proline to leucine mutation in the protein degradation domain and therefore exhibit BR constitutive phenotypes [24,25,29]. However, the *bzr1-1D* and *bes1-D* are morphologically different, indicating the two proteins are involved in distinct BR functions. A number of transcription factors and histone modifiers are identified to interact with BZR1/BES1 for the control of various BR responses [28,30,31,32,33]. BZR1 and BES1 belong to a six-member small family clade, consisting another four homologs, BES1/BZR1 homolog 1 to 4 (BEH1-4), which also act as downstream transcription factors in BR signaling pathway [28].

BR regulates a broad range of plant growth and development, including hypocotyl elongation, root development, skotomorphogenesis, vascular differentiation, floral transition, anther development, and pollen grain maturation. In this review, we will focus on the functions of BRs in reproduction.

## 2. *Arabidopsis* Reproductive Development

*Arabidopsis* reproduction starts from the floral transition, in which the shoot apical meristem (SAM) is transformed into inflorescence meristem (IM), which develops into main stem and flowers. Cauline leaves and axillary branches are produced on the main stem. The floral organ initiates from specific founder cells from IM to form floral primordia and develops into four whorls of floral organs, namely sepals, petals, stamens and carpels from outside to inside. The beginning of floral transition occurs as the florigen accumulates in SAM. The major florigen in *Arabidopsis* is identified as FLOWERING LOCUS T (FT), which is synthesized in the leaf phloem and transported to the SAM by endoplasmic reticulum membrane localized FT-INTERACTING PROTEIN 1 (FTIP1) and several other proteins [34,35]. *FT* and its homolog *TWIN SISTER OF FT* (*TSF*) are exclusively activated by CONSTANS (CO) and GIGANTEA (GI) photoperiodically under light [36,37,38,39,40]. *CO* expression is subject to the circadian clock and its protein stability is degraded by CONSTITUTIVE PHOTOMORPHOGENESIS 1 (COP1) in dark, causing the fast accumulation of FT under long-day conditions [41,42,43]. Therefore, CO, GI and FT are referred to as photoperiod pathway components to induce flowering. In SAM, FT forms a heterodimer with a bZIP transcription factor FD and this complex initiates the transcription of another floral promoter gene *SUPPRESSOR OF OVEREXPRESSION OF CONSTANS 1* (*SOC1*) and the floral meristem-identity gene *APETALA 1* (*AP1*), which promote the formation of floral meristems [39,44,45,46].

*FT* expression is repressed by the key floral repressor FLOWERING LOCUS C (FLC) during vegetative growth. FLC directly interacts with EMBRYONIC FLOWER1-PcG complex to deposit the repressive histone mark, H3K27me3, at *FT* chromatin to antagonize the activation of CO [47]. In addition to the photoperiod pathway, which is activated by exogenous photoperiodic cues, plants also have an endogenous pathway to promote flowering, the autonomous pathway. Components of autonomous pathway are a group of proteins with distinct molecular function but repress *FLC* expression constitutively, such as FLOWERING LOCUS D (FLD), FCA, FY, FPA, and FLOWERING LOCUS K HOMOLOGY DOMAIN (FLK) [48]. In winter-annual *Arabidopsis* ecotypes, *FLC* expression is elevated by the plant-specific coiled-coil protein FRIGIDA (FRI), which forms a stable core protein complex with FRIGIDA LIKE 1 (FRL1), FLC EXPRESSOR (FLX), SUPPRESSOR OF FRIGIDA 4 (SUF4), and FRIGIDA ESSENTIAL 1 (FES1) before vernalization [49,50]. The activity of FRI also depends on the presence of a few active chromatin modifiers and co-transcriptional pre-mRNA modification factors, and these proteins, together with the core FRI complex, form a supercomplex at the *FLC* locus to establish a conducive chromatin spatial structure for high-efficient *FLC* mRNA production [51]. Prolonged cold turns off *FLC* expression and enables flowering in the coming spring (Figure 1A).

Floral organ primordia differentiate from the floral meristem. During floral organogenesis, boundaries are formed by cells with retarded growth rates between whorls of specific organs and each individual flower. Similar boundaries occur between the main stem and lateral branches and cauline leaves. *CUP SHAPED COTYLEDON* (*CUC*) genes are important regulators for both floral organogenesis and lateral organ formation [52,53]. Further, *cuc* mutants show both fused lateral and floral organs, suggesting that *CUCs* are responsible to repress boundary cell growth [54,55].

Proper boundary formation separates flowers into four whorls of organs (sepals, petals, stamens and carpels). In the classic ABC model, sepals are delineated by A genes, and A and B specify petals; in addition, B and C are responsible for stamen development. C gene alone determines carpels and terminates the activities of floral meristem [56]. The MADS box protein AGAMOUS (AG) plays a C role by suppressing the expression of stem cell identity gene *WUSCHEL* (*WUS*) and activates *SPOROCYTELESS/NOZZLE* (*SPL/NZZ*) for microsporogenesis at flower development stage 6 [57,58,59]. In early floral development stage, *AG* expression is activated by WUS and subsequently elevated *AG* expression leads to a repression of *WUS* expression. This forms a WUS-AG feedback loop to turn off floral meristem activity. AG recognizes *WUS* chromatin region and represses its expression by recruiting PcG proteins to deposit the repressive histone mark H3K27me3 [60]. AG also activates the expression of the *WUS* repressor *KNUCKLES* (*KNU*) via competitive binding to the promoter of *KNU* with PcG proteins to indirectly shut down *WUS* expression [61].

*Arabidopsis* floral development can be divided into 20 stages. *AG* expression starts from Stage 3 to diminish *WUS* and initiate floral organ primordia and lasts to very late stages to stimulate and maintain *SPL* expression for the anther development [59,62,63]. *SPL* encodes a transcription factor and functions in ovule formation and early microsporogenesis. *spl* mutant is male sterile owning to the failure of pollen grain development. *SPL* expresses in the anther primordia to establish the microspore mother cells. Morphologically, anther development can be separated into two phases, defined by the completion of meiosis and appearance of microspore [64]. During phase one, sporogenous cells are observed once the four-lobed anther pattern generated and further develop into microspore mother cells which produce tetrads of haploid microspores through meiosis. These microspores are released into the anther locule to begin microgametogenesis in phase two. Finally, matured pollen grains are released by anther dehiscence [64,65]. During phase one, the anther has also developed several highly specialized cells and tissues, including the epidermis, endothecium, tapetum, vascular bundle, and stomium. Each of these cell types and tissues carries out specific functions. For example, the stomium are involved in dehiscence, and tapetum plays a key role in pollen wall formation. The tapetum forms a cell layer surrounding developing microspores within the anther locule. During microspore development, the tapetum provides necessary structural components and nutrients. As the pollen matures, the tapetum diminishes through programmed cell death (PCD) and the released remnants incorporate into the coat of pollen grains [66].

## 3. Role of BR in Floral Transition

BR regulates floral transition in a complicated manner. BR mutants, like *det2-1* and *cpd* have a prominent delay in days to flowering with a slow rate of leaf initiation. Therefore, BR was previously thought to promote flowering [67,68]. However, the IM is transformed from SAM, which produces leaves during vegetative growth. So, the developmental criterion of floral transition is measured by primary leaf number rather than growth rate or days to flowering in *Arabidopsis* [69]. By using this developmental criterion, it is found that *bri1* and BR biosynthetic mutants *det2*, *cpd* and *dwf4* are early flowering [70]. In addition, exogenous application of BR or the steroids, androstenedione and androsterone, cause late flowering in wild type *Arabidopsis* plants, suggesting that BR signaling pathway performs a repressive role in floral transition [70,71]. In *Arabidopsis*, transcription of the central floral repressor *FLC* and its three homologs, *FLOWERING LOCUS M* (*FLM*), *MADS AFFECTING FLOWERING 4* (*MAF4*) and *MAF5*, are depressed in BR mutants and genetic studies demonstrate that BR antagonizes the autonomous pathway by constitutively activating *FLC* expression during vegetative growth [70].

BR activation of *FLC* transcription is mainly through the downstream transcription factors BZR1 and BES1-INTERACTING MYC-LIKEs (BIMs). The dominant *bzr1-1D* mutant exhibits late flowering and defects in BR downstream transcription factor BIMs lead to early floral transition. *FLC* and its homologs are found to be activated in *bzr1-1D* and repressed in *bim1 bim2 bim3*, respectively [24,28,70]. Both BZR1 and BIM1 can recognize and bind to a *cis*-regulatory BRRE element (CGTGTG) located in the first intron of *FLC* and deletion of this *cis*-element abolishes the late flowering phenotype of *bzr1-1D* mutant, suggesting that BZR1 and BIM1 binding to this element is required for *FLC* upregulation. As a transcription factor, BZR1 further recruits the H3K27 demethylase EARLY FLOWERING 6 (ELF6) to the *FLC* region to eliminate the transcription repressive mark H3K27me3, leading to a reduction of H3K27me3 and thus the *FLC* activation [32,70,72] (Figure 1B). Recent studies found that a Cold Memory Element (CME) exists about 0.5Kb upstream of the BRRE *cis*-element in *FLC* first intron, which recruits the B3 domain proteins VP1/ABI3-LIKE 1 (VAL1) and VAL2 [73,74]. VALs bind to the CME before and after vernalization and recruit PcG proteins to add H3K27me3 at the *FLC* locus for transcriptional repression, especially for the maintenance of *FLC* repression after vernalization [74]. Thus, BR signaling may integrate environmental cues to regulate floral transition at a proper time through controlling the level of H3K27me3 at *FLC*.

However, BR seems to participate in floral transition in both *FLC*-direct and -indirect ways in different accessions. For example, BZR1 can also bind to a BRRE *cis*-element (CGTGGG) at *FLD* promoter region and suppresses the transcription of *FLD* [75]. FLD is a component of the autonomous pathway and represses *FLC* expression by reducing the transcription active marks histone acetylation and H3K4me3 levels at *FLC* [76,77]. Therefore, BZR1 can also indirectly activate *FLC* expression through the autonomous pathway. Unlike *bzr1-1D*, the other BR constitutive mutant *bes1-D* flowers like wild type and overexpression of *BES1-D* exhibits no distinct flowering phenotype in Columbia ecotype [70]. While, besides the canonical *BES1* isoform, *BES1* has a long isoform (*BES1-L*) due to alternative splicing [78]. In transgenic plants, overexpressed *BES1-L* causes an early flowering phenotype due to its indirect activation of *FT* [78,79]. BES1 doesn’t bind to *FT* chromatin, but activates *FT* expression through BR ENHAHCED EXPRESSION 1 (BEE1). BES1 binds to *BEE1* promoter to activate *BEE1* expression and *FT* expression can be elevated via BEE1 binding to its promoter region [79,80]. BEE1 activity and protein stability is blue-light dependent, therefore, BR may coordinate with light signaling in the control of the floral transition. Similarly, two P450 enzymes, PHYB ACTIVATION TAGGED SUPPRESSOR 1 (BAS1) and SUPPRESSOR OF PHYB-4 7 (SOB7), are involved in both light response and BR-related floral transition, indicating the complicated regulation of BR on flowering transition [81,82]. In summary, BR plays a negative role in floral transition mainly through direct activation of *FLC* and its homologs, but BR signaling integrates with environmental cues to fine-tune the time of flowering through *FT* and the other flowering pathway.

## 4. Role of BR in Lateral Organ Boundary Formation in Inflorescence

As mentioned above, lowed *FLC* level de-represses *FT* expression, which subsequently activates the expression of *LEAFY* and *AP1* and leads to the transition of SAM to IM [47,48]. BR constitutive and deficient mutants all have organ boundary defects in inflorescence stem and flower organs. *Arabidopsis* wild type plants have straight inflorescence stem, whereas the inflorescence stem of *bzr1-1D* binds toward the lateral organs, such as cauline leaf and axillary branch. In wild type, the axillary branch completely separates from both cauline leaf and the stem, while the axillary branch and cauline leaf are fused together and this fusion causes the inward stem binding in *bzr1-1D* mutant [83]. Similar organ boundary fusion phenotype is also observed in the transgenic lines of *bzr1-1D-CFP*, *BZR1-S173A* and *bes1-D* mutant, where BR signaling is constitutively activated [26,83,84]. This organ fusion phenotype is also observed in *DWF4* overexpression transgenic plants [83]. In contrast, BR deficient mutants, like *bri1-5*, *bin2-1* and *det2-1*, all show the outward binding of IM away from the lateral organs, which is due to the deeper notch between the IM and axillary branch [83]. In addition, the stamen to carpel and stamen to stamen fusions are also observed in *bzr1-1D* and *bes1-D* mutants as well as BR treated wild type flowers, demonstrating that active BR signaling negatively regulates boundary formation [84]. Meanwhile, the extra ovaries occur in flowers of *bri1-5* and *det2-1* is due to the ectopic formation of a boundary structure in the ovary [83,84]. Altogether, BR spatial and temporal accumulation is crucial for organ boundary formation during floral organogenesis.

The organ boundary is formed by the unequal cell division and elongation rates at central meristem and periphery of organ primordia, which separates the lateral organs from central meristem [85]. At the periphery of primordia, cell division and elongation become arrested, whereas cells in lateral organ primordia undergo rapid division and elongation to form leaves and flowers. Unregulated cell division and expansion at organ boundary causes the organ fusion, while insufficient organ boundary formation is the main reason for the outward stem binding in BR deficient mutants [83]. Therefore, the spatial and temporal distribution of BR is crucial for the inflorescence and flower organ formation. Indeed, BR is unevenly distributed in lateral organs, resulting in the dotted localization pattern of BZR1 in the meristem and organ primordia, but not the lateral boundary regions [83,86]. This distinct BR distribution in lateral organs is mediated by a plant specific LOB domain transcription factor LATERAL ORGAN BOUNDARIES (LOB), which exclusively localizes in organ boundaries and represses BR accumulation. Loss of function mutants of *LOB* show fused axillary branch and cauline leaf, and mimic the defects of *bzr1-1D* or BR application [86]. LOB can directly bind to a GCGGCG *cis*-element (LBD motif) at the promoter region of *BAS1* and BAS1 converts bioactive BR to its C-26 hydroxylated derivatives in boundary regions [86,87,88]. Moreover, *LOB* expression is positively regulated by BR in boundary regions, which forms a feedback loop to maintain low level of BR in organ boundary [86].

The accumulated activated form of *BZR1* in inflorescence meristem and sepal primordia represses the expression of organ boundary identity genes *CUC1*, *CUC2*, *CUC3* and *LATERAL ORGAN FUSION1* (*LOF1*) [83,89,90]. Among these genes, the promoter regions of *CUC1*, *CUC2* and *CUC3* contain the BRRE *cis*-elements and can be directly recognized by BZR1 [83]. In addition, proper *CUCs* expression is also regulated by a BES1-TPL repressive module. BES1 recruits the general repressor TOPLESS (TPL) through its C-terminal ERF-associated amphiphilic repression (EAR) motif [84]. Like *bzr1-1D*, *bes1-D* has fused lateral organ defects and BES1 can also repress *CUC3* expression via binding to the same BRRE *cis*-element in the *CUC3* promoter region. Overexpressing *BES1-D* with mutated EAR motif shows normal lateral organ structure, while extra *TPL* expression causes the organ fusion in wild type with depressed *CUC3* expression level [84]. Therefore, it is likely that the BR transcription factor BES1 recruits TPL to repress *CUC3* expression [84]. Genome-wide analysis pointed out that organ boundary specific transcripts overlap with BR repressed genes, and therefore BZR1 and BES1 can regulate a broad range of boundary specific transcriptomes for correct organ boundary development [27,30,90].

## 5. Role of BR in Male Fertility

BR plays a positive role to regulate male fertility by promoting filament elongation and anther development [91]. *Arabidopsis* BR defective mutants in both BR biosynthetic and signaling transduction pathways, like *bri1*, *bin2*, *cpd,* and *dwrf4*, show reduced fertility or male sterility with short filaments and decreased pollen grain number and disposal efficiency [3,67,92,93]. Further, *bri1-116* and *cpd* produce only about 20% pollen grains per anther compared to wild type anther and the relatively small amount of pollens are difficult to release from the anther locule [91]. In BR mutants, the tapetal cells are more vacuolated and enlarged and the number of microspore mother cells are fewer than wild type during early anther developmental stages [91]. During late anther developmental stages, microspores are also degenerated and vacuolated greatly in the mutants. In general, the abnormal development of tapetum, microspore mother cell and microspore all contribute to the decreased number of pollen grains. In *cpd* and *bri1-116* mutants, the pollen grains are hardly released from the anther owing to the absence of bacula/tectum structure in the pollen grain surface [91] (Figure 2A). The defects of pollen grain outer wall in *bri1-116* and *cpd* may be caused by the abnormal development of tapetal cells as the tapetum plays a key role in pollen grain outer surface formation in *Arabidopsis* [94,95].

During early anther development stages, microspore mother cells secret a small peptide ligand TAPETUM DETERMINANT 1 (TPD1), which is perceived by receptor-like kinases Excess Microsporocytes 1 (EMS1) and Somatic Embryogenesis Receptor-Like Kinase 1,2 (SERK1/2) [96,97,98,99]. This signal initiates and maintains the development of tapetum in L2-originated cells [100]. Both EMS1 and BRI1 belong to the large LRR-RLK family and the intracellular domains of BRI1 and EMS1 are functionally exchangeable [101]. Upon perception of TPD1, EMS1 can activate BZR1 and BES1 for the transcription of tapetal development genes [102]. The tapetum failure of *ems1*, *tpd1,* and *serk1* and *serk2* can be rescued genetically by *bzr1-1D* and *bes1-D*. Therefore, BR incorporates with the EMS1-TPD1 signaling for the anther development [101] (Figure 2B).

At molecular level, anther development genes, like *SPL*, *DEFECTIVE IN MERISTEM DEVELOPMENT AND FUNCTION 1* (*TDF1*), *ABORTED MICROSPORES* (*AMS*), *MYB103*, *MALE STERILITY 1* (*MS1*), *MS2* as well as the target genes of MS1 are all depressed in *bri1-116* and *cpd*, consistent with the multiple anther development defects in the mutants. In contrast, transcription of these genes is activated in the *BES1* overexpression lines [91]. The promoter regions of *SPL*, *TDF1*, *MS1*, *MS2*, and *MYB103* contain both E-box and BRRE *cis*-elements and can be recognized and bound by BES1, which demonstrates the direct regulation of BR signaling pathway in anther development [91]. Moreover, the expression of BR receptor *BRI1* is moderately upregulated in *spl* mutant, indicating a feedback regulatory network in BR signaling and anther development [103] (Figure 2B).

However, by using the loss-of-function quintuple and hextuple mutants of *BZR1* family members, it is found that BR downstream transcription factors regulate anther development in both BRI1-dependent and -independent ways [102,104]. Qunituple mutant of *bes1-1 bzr1-1 beh1-1 beh3-1 beh4-1* is phenotypically like wild type but male-sterile and the hextuple mutants of all the six family members are phenotypically similar to the BR insensitive *bri1 brl1 brl3* (*bri1-t*) in both vegetative and reproductive defects [102,104,105]. The anthers of the hextuples are failed to open with no pollen grains. In anther development stage 1, only the L1 layer are developed in *bzr-h*, compared with clearly visible three layers of L1, L2 and L3 in both wild type and *bri1-t*. In stage 2, anther cells of *bzr-h* become severely vacuolated and archesporial cell formation is inhibited in stage 3, which causes the loculeless anther of *bzr-h* [104]. The sporogenesis and archesporial cell initiation are directly regulated by AG and SPL during early male gametogenesis and the expression of the *AG* and *SPL* is almost inhibited in *bzr-h* unopened flowers [59,63,104]. While, *AG* expression is unaffected and *SPL* transcription is only reduced to about half in *bri1-t* [104]. Meanwhile, transcription of tapetal development genes like *DYSFUNCTIONAL TAPETUM*1 (*DYT1*) and *ARABIDOPSIS THALIANA ANTHER 7* (*ATA7*) is largely diminished in the quintuple mutant [104,106,107]. In summary, BZR1 and its five homologs regulate the early stage of anther development through activating the expression of *AG* and *SPL* as well as tapetal development genes in both BR-dependent and independent manners.

In addition to BZR1 and BES1, BIM1 is also involved in another development at early stages. Loss-of-function *BIM1* mutant has mild male sterility at the first few flowers in the main inflorescence due to short stamen filament. This phenotype is similar to *squamosa promoter binding protein-like 8* (*spl8*) mutant and is enhanced in the double mutant of *bim1 spl8* [108]. *SPL8* encodes an SBP DNA binding protein and expresses highly at early anther development stages. Further, *spl8* is semi-sterile with short filament and failure of microspore mother cell initiation in the first twelve flowers [109]. In the double mutant, microspore mother cells fail to produce in the adaxial loculi at anther development stage 5 and the initiated microspore mother cells cannot undergo meiosis at stage 6, resulting in the disappearance of pollen sacs in the anther and male sterile. *BIM1* has a similar expression pattern with *SPL8* in early developing anther and they function cooperatively in anther development [108]. Moreover, BIM1 contributes to embryonic patterning via physical interaction with DORNROESCHEN (DRN) and DORNROESCHEN-LIKE (DRNL) [110]. In embryonic globular stage, the asymmetrical cell division is seldomly occurred in *bim1*, leading to the absence of central and/or basal domain in embryo and thus defects in cotyledon formation. This abnormal embryonic patterning of *bim1* is proportionally increased in the double mutants of both *bim1 drn* and *bim1 drnl* [110]. DRN and DRNL regulate auxin-related embryonic patterning by affecting PIN-FORMED 1 (PIN1) localization [110]. The protein interaction of BIM1 with DRN and DRNL may give hint to the crosstalk of BR and auxin in embryonic patterning.

## 6. Role BR in Other Plant species

Like in *Arabidopsis*, BR plays similar effect on reproduction in other plants species. In monocotyledonous wheat, exogenous application of 24-epibrassinolide delays vegetative to generative transition, while the BR inhibitor, Brassinazole, accelerates the transition and heading stage [111]. In short-day plant *Pharbitis nil*, treatment of BL and CS inhibit flowering in combination with the inductive photoperiod, suggesting that BR integrates with environmental cues for the proper reproductive transition [112]. In addition to floral transition, BR also regulates both male and female fertility in crops. In maize, ineffective BR biosynthesis causes male sterility, which is due to the failed anther and pollen development [113,114]. In bisexual flower fruit cucumber, exogenous BR application accelerates and increases female flower organogenesis in a dose-dependent manner and this effect may attributes to the crosstalk of BR-ethylene [115].

## 7. Conclusions

In summary, BR plays a broad role in plant reproduction from floral transition to male fertility. Plant development is modulated by plant-environment crosstalk that requires the incorporation of external environmental cues and internal responses. Studies have revealed the crucial roles of hormone in plant development and plant-environment interaction. As mentioned above, the effects of BR signaling are also subject to hormone crosstalk and biotic as well as abiotic stimuli. For example, BR-mediated cell elongation is positively affected by auxin and gibberellin but negatively by cytokinin and abscisic acid. During reproduction, the cell identity is established within a few cells at the initiation of floral transition and lateral organ formation. Unlike auxin, BRs do not undergo distant transportation and are believed to exert their functions locally [116]. All these call for the precise temporal and spatial control of BR signaling in reproductive development. The application of effective techniques to isolate tissue-specific plant cells and to characterize tissue/cell specific transcriptome and/or proteome is becoming available, such as Isolation of Nuclei TAgged in specific Cell Types (INTACT), cleavage under targets and release using nuclease (CUT & RUN) and single cell sequencing. All these will facilitate our understanding of the function and mechanism of BR signaling in the specific development process as well as a response to environmental stimuli [117,118,119].

## Figures and Tables

**Figure 1 ijms-21-00872-f001:**
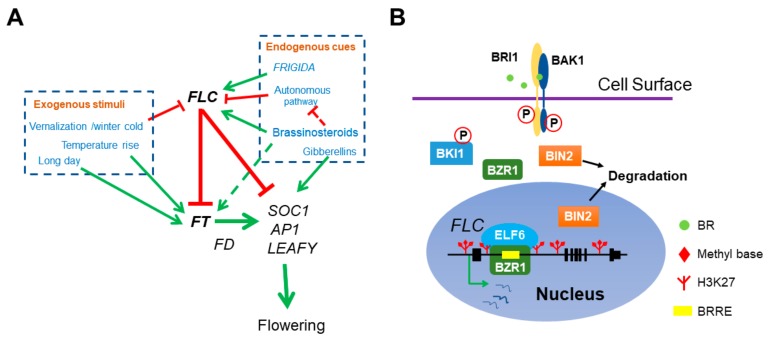
The role of BR in *Arabidopsis* flowering network. (**A**) BR directly represses floral transition by activating the transcription of main floral repressor *FLC*. Both endogenous cues and exogenous stimuli can affect flowering time. FRI and BR delay flowering by elevating *FLC* expression and the autonomous pathway genes antagonize with FRI and BR by constitutively repressing *FLC*. Meanwhile, vernalization represses *FLC* and ensures flowering after plants return to warm. FLC interacts with EMF-PRC2 repressive complex to repress the transcription of florigen *FT* and *SOC1*. The ambient temperatures and photoperiod pathway activate *FT* expression. FT associates with another transcription factor FD to stimulate the transcription of *SOC1*, *AP1* and *LEAFY* and thus initiates the transition of SAM to IM. In addition, BR also indirectly mediates flowering time by activating *FT* and repressing the autonomous pathway member *FLD* expression (dotted lines). (**B**) The molecular mechanism of BZR1 activation on *FLC* expression. BZR1 recognizes a BRRE *cis*-element in *FLC* 1st intron and recruits the histone demethylase ELF6 to erase the repressive histone mark H3K27me3 at *FLC* locus.

**Figure 2 ijms-21-00872-f002:**
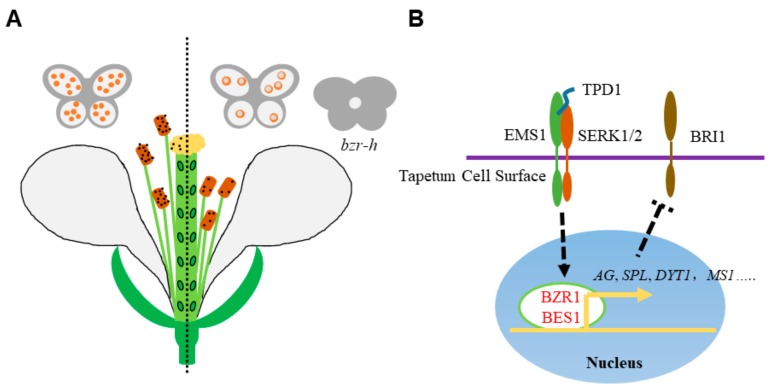
BR promotes the male reproductive development. (**A**) The schematic drawing of male reproductive defects in BR-deficient mutants. The left part shows the wild type normal male reproductive organ development. In the BR-deficient mutants (right part), the stigma elongation is retarded, the anther is failed to release pollen grains and the microspores are vacuolated and degenerated. Meanwhile, loculeless anther is produced in the hextuple mutant *bzr-h*, indicating the BR downstream transcription factor BZR1 and its homologs play both BR-dependent and -independent role in anther development. (**B**) Roles of BZR1 and BES1 in anther development. In tapetum cells, the cell membrane localized receptor kinases EMS1 and SERK1/2 perceive the peptide ligand TPD1 and activate BZR1 and BES1. BZR1 and BES1 then initiate the transcription of anther development genes, like *AG*, *SPL*, *DYT1*, *MS1* and so on. The expression of *BRI1* is elevated in *spl* mutant, suggesting a possible regulatory network of anther development and BR signaling. Dotted arrow indicates the activation of BES1 and BZR1 by EMS1-TPD1-Serks signaling. Dotted T bar represents for the feedback downregulation of *BRI1* transcription by SPL.

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
