# Peer review of "Roles of Brassinosteroids in Plant Reproduction"

_ijms, 2020, doi:10.3390/ijms21030872_

Round 1

Reviewer 1 Report

This manuscript is clearly conclude many aspects of the relation between BR and flower development. I have a few recommendations as following;

L62: BRRE should be first mention as "BR-Response Element" or include in lists of abbreviation.

L65-66: "The brz1-1D and......BR functions." I disagree this sentence. Phenotype of brz1-1D and bes1-D is quite different. brz1-1D showed the downward curling leaves but bes1-D is not. bes1-D showed long petioles and some of petioles and leaf blades were curved. 

L184-194 and figure 1: Authors have lots of detail of BR function on flower development. Some function such as BRZ1 and BES1 are not exactly same. It is better if authors give more detail in figure 1.

Lastly, since this review talk all aspects of BR on plant reproduction, I think "the promotion of female flower in Cucurbit by BR" should be described.

Author Response

This manuscript is clearly conclude many aspects of the relation between BR and flower development. I have a few recommendations as following;

Response:

We thank the reviewer for the positive comments.

L62: BRRE should be first mention as "BR-Response Element" or include in lists of Abbreviation.

Response:

We have added BR-Response Element in the main text (Line 57 and 58) as well as in the Abbreviations part.

L65-66: "The brz1-1D and......BR functions." I disagree this sentence. Phenotype of brz1-1D and bes1-D is quite different. brz1-1D showed the downward curling leaves but bes1-D is not. bes1-D showed long petioles and some of petioles and leaf blades were curved. 

Response:

We are sorry for this inaccurate description, and have revised the sentence as indicated in line 60.

L184-194 and figure 1: Authors have lots of detail of BR function on flower development. Some function such as BRZ1 and BES1 are not exactly same. It is better if authors give more detail in figure 1.

Response:

Since BR plays a negative role in floral transition not only in dicot Arabidopsis but also in monocot wheat, we added the molecular mechanism of BZR1 activation of FLC expression in Arabidopsis in Figure 1.

Lastly, since this review talk all aspects of BR on plant reproduction, I think "the promotion of female flower in Cucurbit by BR" should be described.

Response:

We have changed the subtitle “Role of BR in male fertility” to “Role of in fertility” and added the function of BR in Cucurbit female flower in line 254-256.

Reviewer 2 Report

Article is devoted to role of plant steroid hormones - brassinosteroids (BR) in plant generative development. Since there is lot of articles about BR such review is always useful thing. Article is  written rather good but I have few suggestions:

Abstract should be more focused on plant generative development than on other problems. Now, first 3 sentences are about mechanism of BR binding (and it takes almost half of Abstract text). Article is mainly about molecular mechanisms of BR regulation of reproduction,  may be authors should just underline this aspect in Abstract. Article describes BR regulation of development mainly for Arabidopsis. May be would be good also mention about fact that exogenous BR in concentration dependent manner rather delayed flowering of this plant.

Janeczko A., Filek Wł. Biesaga-Kościelniak J., Marcińska L, Janeczko Z. 2003. The influence of animal sex hormones on the induction of flowering in Arabidopsis thaliana: comparison with the effect of 24-epibrassinolide. Plant Cell Tissue and Organ Culture. 72 (2): 147-151

In my opinion authors also should mention about BR impact on development of other plants like cereals (maize, wheat).

 Makarevitch I., Thompson A., Muehlbauer G.J., Springer N.M. 2012. Brd1 gene in maize encodes a brassinosteroid C-6 oxidase. PLoS ONE 7: e30798.

Janeczko A., Oklestkova J., Novak O., Śniegowska-Świerk K., Snaczke Z., Pociecha E. 2015. Disturbances in production of progesterone and their implications in plant studies. Steroids 96: 153–163.

Author Response

Article is devoted to role of plant steroid hormones - brassinosteroids (BR) in plant generative development. Since there is lot of articles about BR such review is always useful thing. Article is  written rather good but I have few suggestions:

Response:

We thank the reviewer for these positive comments.

Abstract should be more focused on plant generative development than on other problems. Now, first 3 sentences are about mechanism of BR binding (and it takes almost half of Abstract text). Article is mainly about molecular mechanisms of BR regulation of reproduction,  may be authors should just underline this aspect in Abstract.

Response:

We deleted the second and third sentences and emphasized the molecular mechanism in line 17 and 18.

Article describes BR regulation of development mainly for Arabidopsis. May be would be good also mention about fact that exogenous BR in concentration dependent manner rather delayed flowering of this plant.

Janeczko A., Filek Wł. Biesaga-Kościelniak J., Marcińska L, Janeczko Z. 2003. The influence of animal sex hormones on the induction of flowering in Arabidopsis thaliana: comparison with the effect of 24-epibrassinolide. Plant Cell Tissue and Organ Culture. 72 (2): 147-151

Response:

We thank for pointing out the missing reference and it has been added in line 161-163.

In my opinion authors also should mention about BR impact on development of other plants like cereals (maize, wheat).

 Makarevitch I., Thompson A., Muehlbauer G.J., Springer N.M. 2012. Brd1 gene in maize encodes a brassinosteroid C-6 oxidase. PLoS ONE 7: e30798.

Janeczko A., Oklestkova J., Novak O., Śniegowska-Świerk K., Snaczke Z., Pociecha E. 2015. Disturbances in production of progesterone and their implications in plant studies. Steroids 96: 153–163.

Response:

We have added the wheat work in part 3. “Role of BRs in floral transition” line 163-164 and the maize work in part 5. “Role of BR in fertility” in line 257-258.